# Dairy By-Products: A Review on the Valorization of Whey and Second Cheese Whey

**DOI:** 10.3390/foods10051067

**Published:** 2021-05-12

**Authors:** Arona Figueroa Pires, Natalí Garcia Marnotes, Olga Díaz Rubio, Angel Cobos Garcia, Carlos Dias Pereira

**Affiliations:** 1Polytechnic Institute of Coimbra, College of Agriculture, Bencanta, 3045-601 Coimbra, Portugal; arona@esac.pt (A.F.P.); natali@esac.pt (N.G.M.); 2Department of Analytical Chemistry, Nutrition and Bromatology, Faculty of Sciences of Lugo, Food Technology Area, University of Santiago de Compostela, E-27002 Lugo, Spain; olga.diaz.rubio@usc.es (O.D.R.); angel.cobos@usc.es (A.C.G.); 3Research Centre for Natural Resources, Environment and Society (CERNAS), Bencanta, 3045-601 Coimbra, Portugal

**Keywords:** valorization, cheese whey, second whey cheese, ultrafiltration

## Abstract

The search for new food products that promote consumers health has always been of great interest. The dairy industry is perhaps the best example regarding the emergence of new products with claimed health benefits. Cheese whey (CW), the by-product resulting from cheese production, and second cheese whey (SCW), which is the by-product of whey cheese manufacture, have proven to contain potential ingredients for the development of food products with improved nutritional characteristics and other functionalities. Nowadays, due to their nutritional quality, whey products have gained a prominent position among healthy food products. However, for a long time, CW and SCW were usually treated as waste or as animal feed. Due to their high organic content, these by-products can cause serious environmental problems if discarded without appropriate treatment. Small and medium size dairy companies do not have the equipment and structure to process whey and second cheese whey. In these cases, generally, they are used for animal feed or discarded without an appropriate treatment, being the cause of several constraints. There are several studies regarding CW valorization and there is a wide range of whey products in the market. However, in the case of SCW, there remains a lack of studies regarding its nutritional and functional properties, as well as ways to reuse this by-product in order to create economic value and reduce environmental impacts associated to its disposal.

## 1. Introduction

Despite the controversy about the impact of dairy products on health, the consumption of dairy products in the world is increasing steadily. The development of new dairy products containing prebiotics and probiotics is also increasing based on their benefits for human health. Probiotic foods contain microorganisms that have therapeutic properties like antimicrobial activity, hypocholesterolemic activity, maintenance of gastrointestinal balance and anticarcinogenic activity [1]. Industry has developed a large group of new, nutritionally improved products, which have been a success on the market. Therefore, traditional dairy products have changed and dairy industries need to evolve into the new generation of dairy products with differentiated characteristics regarding health and nutritional properties [2,3].

Due to the increase in food consumption and to the stringent environmental regulations, the management of food waste and by-products is a challenge for the agri-food industries that face demanding economic costs for their treatment and/or disposal. The dairy industry annually produces millions of tons of by-products, the main component of which is cheese whey (CW), which corresponds to the net fraction remaining after milk coagulation. Around 9–10 L of whey results from the production of 1 kg of cheese and if discarded without treatment it creates a significant problem for the environment [4,5].

CW has a high Biochemical Oxygen Demand (BOD) and a high Chemical Oxygen Demand (COD) [6]. When discarded into water sources, it reduces the dissolved oxygen, and poses a major risk to aquatic life, as well as to the environment and human health [7]. As is the case of CW, second cheese whey (SCW) is considered a significant source of pollution, possessing high values of BOD and COD (*ca.* 50 and 80 g L^−1^, respectively) [8]. Lactose (35–50 g L^−1^) is the principal constituent responsible for the high COD values (>70%) [9]. SCW represents a considerable problem because its valorization is not a common practice and it is difficult to manage as animal feed, since most animals are not able to digest high amounts of lactose without suffering from digestive disorders [8,10]. Comparatively, the average BOD and COD values for urban wastewaters are 0.20 and 0.41 g L^−1^, respectively, which represents around 1/150 of the pollution charge of both by products, CW and SCW [11].

CW is nowadays recognized as a source of functional and bioactive compounds, especially proteins and peptides. However, a large proportion of the whey produced worldwide is still not valorized. This results from the fact that small and medium size dairy industries lack dimension to make the necessary investments for CW valorization [4,6].

In some countries such as Portugal, Spain, Italy and Turkey, CW is employed in the production of whey cheeses (Requeijão, Requesón, Ricotta and Lor, respectively) and other products with nutritional and medical potential [3,12,13]. Normally, these products are typically obtained from ovine, caprine, bovine or buffalo cheese whey. CW can be previously acidified, as is the case of Ricotta, followed by heating at temperatures around 85–90 °C for 20–30 min, to allow coagulation and subsequent precipitation of whey proteins and separation of whey cheese mass [1,6,8]. Whey cheese yield is quite variable depending on the origin of the whey and the process employed but, unless whey is previously concentrated, it is lower than 4%. The liquid remaining after whey cheese separation represents more than 90% of the original whey and is called second cheese whey (SCW), Sorelho in Portugal or Scotta in Italy, are the major by-product of whey cheese production. Lactose (4.8–5.0%), salts (1.0–1.13%), and proteins (0.15–0.22%) generally compose SCW resulting from bovine milk [6,9,14]. However, the protein and fat contents of SCW resulting from ovine milk can represent 1–4%.

SCW is a poorly studied by-product and there is little interest in its recovery [6]. Some authors studied the use of SCW for conversion into biofuel and as a biotechnological substrate for fermented products while others studied its potential for the production of fermented drinks [8,14,15,16]. Dried SCW protein concentrate was also evaluated for its usefulness as food ingredient, based on the functional properties of their proteins [17]. However, the available literature and research works concerning SCW are very limited. 

In the present work, special attention will be given to the more recent research regarding the processing and potential applications of SCW, envisaging the reduction of the environmental pollution and the economic valorization of this by-product by developing new products with potential benefits to human health. 

## 2. Cheese By-Products

### 2.1. Whey

CW has a yellow-green color due to the presence of riboflavin and contains about 65 g of total solids per liter. Whey represents 85–95% of the milk volume, retains about 55% of the milk nutrients and approximately 20% of the total protein content [3]. The composition of CW depends on the origin of milk, the types of cheese manufactured (rennet or acid coagulation), and generally, on the factors that affect milk composition such as, breed, seasonal cycles, feed, and lactation phase. 

This by-product can be divided in two types: sweet and acid whey, being the processing technique used that influences the type of whey. Sweet bovine whey has a pH around 6–7 [10,18] 6–10 g L^−1^ protein, 5–6 g L^−1^ fat, 46–52 g L^−1^ lactose and 2.5–4 g L^−1^ minerals and results from the production of most types of cheese or some casein products [3,19]. The first step in the manufacture of cheese (and sweet whey) is addition of rennet to milk. The rennet works by curdling the casein present in the milk leading to the formation of curd. The curd is then strained from the remaining liquid (CW). The rennet induced coagulation of casein occurs at pH 6.0–6.5 [10]. 

Acid whey has a pH of approximately 4.5–5.8 [10,18], 6–8 g L^−1^ protein, 5–6 g L^−1^ fat, 44–46 g L^−1^ lactose and 4.3–7.2 g L^−1^ minerals [19]. This whey results from the activity of lactobacilli or by the addition lactic acid or mineral acids used to coagulate the casein for the manufacture of most types of industrial caseins. It also results from the production of acid curdled cheeses, such as cottage cheese. Lactose content is generally lower in acid whey, but the mineral content normally exceeds that of sweet whey [18,20].

Lactose, the main solid component of CW, representing 70–75% of the total solids, is responsible for the high values for BOD and COD [3,9,21]. World production of bovine whey was estimated around 160 million tons per year [22]. The global production of cheese was expected to reach 21 million tons in 2020. This corresponds to an estimated global production of ca. 168 million tons of cheese whey [23]. About 50% of this whey is considered waste, used as animal feed, biofertilizer in irrigation systems, or discharged without treatment [21,22,24,25]. The conventional solutions for the treatment of whey include the production of dehydrated whey products such as whey powder (WP), whey protein concentrates (WPC), whey protein isolates (WPI) and powdered lactose [21].

Isolated whey proteins also have uses in the food industry due to their physic-chemical and nutritional properties, which allow them to act as emulsifiers, gelling agents, water binders and foaming/whipping agents in food systems. They are used in many different foods including soups, salad dressings, processed meat, dairy and bakery products or specific whey products such as Ricotta or other whey cheeses, as well as in fermented whey drinks [3,21,26,27].

Particular attention has been given to the benefits of CW regarding the nutritional, biological and functional value of whey proteins. More recently great attention has been given to minor components, important in human health, such as whey protein derived bioactive peptides with antihypertensive, antioxidant and antimicrobial activities, and non-digestible oligosaccharides, claimed to behave as dietary fibers and prebiotics [25,28].

The described processes for whey solids concentration include evaporation, ultrafiltration (UF), nanofiltration (NF) or reverse osmosis (RO), before dehydration by spray or freeze drying. However, these processes cannot be applied by small and medium companies as they are expensive and require large installations being the factor that limits the amount of CW valorized for human consumption [24]. Despite the fact that in several European countries, a significant amount of CW is used to obtain whey cheeses by thermal aggregation of the whey proteins only a limited amount of CW is used in the manufacture of such products [13]. 

The composition, as well as the high volumes produced, associated to the environmental impact of CW and SCW, are drivers for the great interest in their valorization. It should be emphasized that ovine SCW contains a protein and mineral content higher than that presented by bovine whey (Table 1). Thus, the appreciation of this by-product for possible future applications is of great interest.

### 2.2. Second Cheese Whey (Sorelho)

SCW is the by-product resulting from whey cheese production (e.g., Requeijão)*,* whose manufacturing process consists basically of heat processing the whey at temperatures around 85–90 °C for 20–30 min aiming at denaturing the whey proteins [8] (Figure 1). In most cases, as happens with CW, small and medium companies are not able to valorize the SCW components for human consumption and this product is used as animal feed. Thus, the high volumes of CW (about 8 L per kg of cheese) and SCW (about 18 L per kg of whey cheese) produced remain a problem for cheese producers as they pose valuation/environmental challenges [32].

In Italy only about 15% of CW obtained annually is used to produce Ricotta cheese generating about 1 million tons of SCW [8]. SCW is also produced in Spain, Portugal, Greece, Turkey and Romania, where it is normally used for animal feed or treated as waste. 

In the past, economical and easy solutions were used by dairy industries to discard SCW such as spreading of the product on fields or elimination in rivers, lakes and ocean [33]. Nowadays, most of SCW is used as supplement feed for livestock. However, this by-product is a natural source with important compounds such as proteins mostly in the denatured state, soluble peptides, oligosaccharides, lactose, non-protein nitrogen, hydro-soluble vitamins, a variety of minerals and free amino acids [34,35]. It can be used as a good substrate in biotechnological processes to produce commercial high-value compounds [14,16,28]. SCW contains approximately 50% of the original dry matter of the whey. Lactose and minerals represent most to its dry mass, but residual fat and non-thermally precipitated nitrogen components are also present. Hence, attempts to recover some of the solid components present before the final disposal may be of interest [36]. The composition of SCW varies widely and that depends on the CW origin and processes employed to produce whey cheese. SCW resulting from bovine whey contains proteins (0.15–0.22%), salts (1.0–1.13%) and lactose (4.8–5.0%). SCW obtained from ovine whey is generally characterized by a higher protein content [6,14] and contains ca. 6.71% dry matter, 0.49% of proteins, 0.53% fat and 2.08% of minerals [15]. 

The mineral content of ovine SCW is quite high due to the addition of salt to the milk during cheese manufacture [9,28,34]. This by-product has acidic characteristics with pH values around 3–6 [9]. High concentrations of nutrients such as ammonium nitrogen (0.06–0.270 g L^−1^) and total phosphorus (0.006–0.5 g L^−1^) are also present in SCW [21].

Therefore, due to its composition SCW can be used for various different purposes including food formulations, nutraceuticals and biofuel products [8,28,34,37,38].

Lactose is an important ingredient in infant formula and in food processing [39]. The content of lactose in SCW is very high and its recovery may be a solution, which combined with the valorization of the nitrogen fraction, can be appealing. For lactose recuperation in CW, generally, the permeate from the UF process containing around 4–8% lactose and 0.5% salt is concentrated to 60% solids in multiple effect evaporators. Lactose is then crystallized from the concentrate, separated, and dried into lactose powder. To enhance the yield and purity, the UF permeate is also often processed by NF to partially remove the salts and to increase the lactose concentration [39]. This process can also be applied to SCW in view of its valorization.

## 3. Whey and Second Cheese Way Nitrogen Compounds

CW presents an important mixture of proteins with chemical, physical and functional properties. These proteins have important roles in nutrition and in specific physiological actions, namely the ability to bind metals or specific activities in the immune or digestive systems and are an important source of essential amino acids [40]. Whey proteins are also rich in branched-chain amino acids (BCAAs), important in muscle health and as metabolic regulators in protein and glucose homeostasis and lipid metabolism (weight control). They also present sulfur-containing AAs that exert metabolic regulation and act as precursors of the potent intracellular antioxidant glutathione. The composition of whey proteins also represent a better source of essential AAs when compared to egg, meat and soy proteins [41,42]. 

Besides, there is an increasing interest in whey as a potential source of bioactive compounds that can reduce the risk of diseases [39,42,43]. 

Whey proteins have a globular structure, with a uniform distribution of non-polar, polar and charged groups. These proteins include β-lactoglobulin (β-LG), α-lactalbumin (α-La), immunoglobulins, serum albumin (SA), lactoferrin, lactoperoxidase and enzymes (approximately 60), in addition to other protein components, such glycomacropeptide or caseinomacropeptide, which is released from κ-casein in the first step of enzymatic curdling [43,44,45,46,47]. These proteins have an amino acid profile different from caseins (a greater amount of sulfur-containing amino groups), are easily denatured by heat and are dephosphorylated [46]. Whey proteins can be used as simple protein supplements or can be used as ingredients in transformed food formulations [18,48].

β-Lg is the major protein in CW [25,47], representing around 50% of the total bovine whey proteins. It has a globular structure with 162 amino acid residues, has a molecular weight (MW) of approximately 18.4 kDa and a pI value of 5.1 [46,49]. It is an important source of essential amino acids and due to its excellent nutritive properties is applied in sport and dietary nutrition [50]. β-Lg is also used due to its emulsification, foaming and gelling properties, being also used in the manufacture of protein hydrolysates for ingredient formulation [39]. 

α-La is the second major protein of whey [47], representing around 20% of bovine whey proteins. It is a calcium metalloprotein composed by 123 amino acids, being a strong Ca^2+^ binding protein. Its MW is about 14.2 kDa and it has a pI value of 4.2 [49]. This protein is more resistant to denaturation by heat than β-LG, in neutral pH, since below 75 °C its denaturation is reversible [46]. Its nutritive properties are very important, being the α-La purified fractions used in infant foods to adjust its protein content close to that of human milk [50]. Other reported properties of α-La are its antihypertensive and antioxidant activities, anti-obesity potential and the anti-tumoral activity observed in the complex between human α-La and oleic acid called BAMLET (human alpha-lactalbumin made lethal to tumor cells) [10,51]. 

Serum Albumin (SA) represents around 6% of bovine whey proteins, has 582 amino acids residues, an MW of 69 kDa and a pI value of around 4.7–4.9. Biological functions of SA include fatty-acid binding and antimutagenic activity. SA has good gelling properties and is commonly used in food and therapeutic applications, for cancer prevention and as an antioxidant [44,49].

Immunoglobulins (Igs) are γ-globulins found in blood and body fluids in all lactating species. All Igs have a basic structural unit that comprises four polypeptide chains with two heavy chains with an MW around 50–70 kDa and two light chains with an MW around 25 kDa, held together with disulfide bonds. These proteins are widely used by the immune system to identify and neutralize bacteria, viruses and other antigens, and have anti-cholesterol, antimicrobial and antiviral properties. Human immunoglobulins are classified in different classes: IgA, IgD, IgE, IgG, and IgM, but bovine milk mainly contains IgG, IgM and IgA [44,49].

Lactoferrin (LF) is a single-chain polypeptide characterized to be an iron-binding glycoprotein also known as lactotransferrin. It has an MW of 80 kDa, is composed of 691 AA residues and has a pI value of 8.6. Lactoferrin represents one of the minor proteins in whey. It is important in iron transport and absorption in young gut, in defense against pathogens and also shows antioxidant, anti-inflammatory and antimicrobial activities [10,46].

Lactoperoxidase (LP) has an MW of around 78 kDa, 612 amino acid residues and a pI value of 9.6. This protein is very important for milk preservation [49] and has antimicrobial, bacteriostatic, bactericidal and antifungal activities [10,46].

Caseinomacropeptide (CMP) is composed by the 64 C-terminal amino acids of κ-casein. It is adequate for special diets, due to its BCAAs (valine and isoleucine). CMP interacts with toxins, viruses and bacteria due to its carbohydrate fraction, preventing the binding of different pathogens to cells. Studies have also demonstrated the CMP protecting effect against acidic erosion of teeth. CMP also exerts immunomodulation and prebiotic activities [51].

The amount and state of whey proteins in foods depend of several factors: whey origin (cheese variety and type of milk), the production process and process conditions used during cheese production and the conditions used for whey powder, whey protein concentrates (WPC) or isolates (WPI) manufacture namely, pH and heat treatment, as well as the characteristics of the food matrix where they are incorporated alongside with the operational conditions to produce it [52]. 

The nutritional value of bovine, ovine and caprine whey proteins (Table 2), and their technological properties have proven distinguished health benefits when hydrolyzed due to the resulting bioactive peptides [7]. Over the last years, scientists focused their studies on bioactivities associated with whey protein-derived peptides [51,53]. These peptides are defined as specific protein fragments who have a positive impact on body functions, a positive influence on physiological and metabolic functions, as well as, antimicrobial, antihypertensive, anticytotoxic, antioxidative, immunomodulatory and mineral-carrying activities. 

The fractionation of whey proteins in peptides can also improve their functional properties, such as solubility, emulsifying power, and texture [10,44,54,55,56].

Natural digestion of whey proteins in the gastrointestinal tract is not enough to promote a positive health effect, thus the bioactive peptides must result from industrial processing of hydrolyzed whey and second cheese whey [18,41]. These processes will be reviewed in the following section.

**Table 2 foods-10-01067-t002:** Relative proportions of bovine, ovine and caprine whey proteins in whey (%).

Whey Proteins	Bovine Whey	Ovine Whey	Caprine Whey
β-Lactoglobulin	53.3–66.0	73.1	46.6
α-Lactalbumin	15.0–20.0	17.9	18.3
Serum Albumin (SA)	6.0–7.0	2.7	5
Immunoglobulins	11.0–13.3	-	-
Lactoferrin	0.7-3.3	1.6	2
Lactoperoxidase	0.5–1.0	-	-
Enzymes	0.5	-	-

Adapted from [17,25,45,51,57].

The main problem impairing the industrial use of CW and SCW is their relatively low concentration in proteins which implicates the use of concentration processes to assure high hydrolysis yields [58].

The development of membrane separation techniques has been essential for the food industry that has taken advantage of their relatively easy scale-up, as well as to the fact of being inexpensive when compared with preparative chromatographic techniques. Membrane separation techniques also offer the advantage that during processing the bioactive compounds do not suffer drastic heat treatments [58,59]. These techniques will be detailed in Section 6. Several methods, such as UF, diafiltration (DF), NF, ion exchange chromatography, electrophoresis, crystallization and precipitation have been used to concentrate and separate proteins from other components present in whey. These techniques have been applied not only to the major compounds (β-lactoglobulin and α-lactalbumin), but also to other minor compounds, such as lactoperoxidase, lactoferrin, immunoglobulins, as well as other products such as the biologically active peptides obtained by enzymatic or chemical modification of proteins [43].

From CW, several types of products can be obtained: whey powder (WP), whey protein concentrates (WPC), whey protein isolates (WPI), whey protein hydrolysates (WPH), delactosed whey, demineralized whey, whey permeate and lactose [21,60].

## 4. Whey Protein Concentrates (WPC), Whey Protein Isolates (WPI) and Whey Protein Hydrolysates (WPH) 

Generally, whey proteins are highly valued and come in many different forms, such as whey powder (8–12% protein and over 70% lactose), WPC, containing between 30–89% of protein and WPI containing over 90% of protein and almost no lactose (around 3%). 

Processing steps in the manufacture of WPC and WPI may sometimes cause some protein denaturation, which tends to affect their functionality [43,61]. In the production of WPC and WPI, the most popular techniques for the pre-concentration of whey are UF and/or DF, as well as some other membrane processes such as NF reverse osmosis (RO), electrodialysis (ED) and microfiltration (MF) [62]. To produce WPC or WPI the UF/DF process is employed and, afterwards, the concentrated whey is pasteurized, evaporated and dehydrated (Figure 2). WPH contain about 80% of proteins that have been hydrolyzed by specific enzymes; therefore, they are more easily metabolized and have nutraceutical properties [10,21,45].

WPC, WPI and WPH are, at the moment, the best ways to valorize whey proteins. The variations in the composition and functional properties of these products result from several factors, such as the source of the milk (bovine, caprine or ovine) and the conditions in which cheeses are produced. However, the main cause for the variations in proteins content and state are the factors related to the obtention of cheese whey (sweet or acid) rather than the process of production of the WPC and WPI [63]. Processing conditions, namely thermal treatments, pH and type and quantity of salts present also play a significant role in the functional properties of these products. Recent works evaluate the impact of thermal treatments and emerging technologies on the structure and techno-functional properties of milk proteins or characterizing the changes of bovine milk serum proteins after simulated industrial processing [64,65].

WPC and WPI have a wide range of food applications due to their high protein content and can function as water-binding, gelling, emulsifying and foaming agents. Benefits of WPC and WPI in food applications include their high protein and valuable amino acid contents, low calorie and low fat contents, good emulsification, foaming and gelling properties and compatibility with other ingredients associated to the perception that it is a “natural” product [45]. However, in Portugal, only half of the bovine whey (ca. 300,000 tons/year) is processed to produce ca. 13,000 ton of whey powder (12% protein). The remaining whey produced is used directly for animal feed [66].

WPC production includes a final drying step carried out by spray or freeze drying. The cost of such equipment is high, which hampers their application in small/medium scale dairy industries [21].

For WPI production, adsorption techniques such as ion exchange chromatography can also be used since it provides an additional level of selectivity on the use of membranes. The product obtained has better functional properties than WPC and its economic value is also higher [67,68].

Some examples for the use of WPC and WPI in foods, such as cheeses, yoghurts, sauces and fermented drinks are referred by several authors [17,26,29,41,63]. WPC can also be incorporated as emulsifiers in salad dressings. Good emulsifying properties were observed, and the salad dressings showed high firmness and stability [63].

For WPH production, the digestive enzymes trypsin, pepsin, and chymotrypsin, plant enzymes mainly papain, bromelain and cardoon enzymes, as well as bacterial proteases namely originated from *Bacillus licheniformis*, and *Bacillus subtilis* or mixtures of some of these enzymes [58,60] are used in specific conditions of temperature and pH that favor the hydrolysis process. WPH are soluble in a wide range of pH, have good viscosity in solution due to water binding, promote cohesion, adhesion, elasticity and improve the emulsification of fat and whipping. WPH can also form flexible edible films, as is the case of WPC or WPI. WPH are good emulsifiers and when used in addition with polysaccharides their emulsification ability is improved. Their functional and biological properties depend on the type of enzyme used in terms of specificity and selectivity, the hydrolysis conditions (enzyme-to-substrate ratio, incubation temperature, pH, time) and the source of the original protein, native or denatured [60].

Enzymatic hydrolysis of proteins allows the selection of the protein substrate and enzyme specificity to optimize the yield of bioactive peptides [18,43]. WPH contain peptides which increase the digestibility, bioactivity and nutritional properties of foods to which they are added and can be used in many different applications, such as gelling agents, emulsifiers or foaming agents, depending on the nature of the peptides produced and the degree of protein hydrolysis [69]. Table 3 presents the sequences of the bioactive peptides observed in WPC hydrolysate obtained using *Cynara cardunculus* L. extracts. Most of the peptides derived from α-La, but some of them resulted from CMP, β-Lg or β-Casein (β-Cas) [46]. 

Many favorable properties for the human health have been claimed for some of these peptides. The cardiovascular effects [58] opioid, antithrombotic and antioxidant activities [55] are some examples. Several studies confirm the applicability of health-promoting peptides in the production of functional foodstuffs. The hypotensive effect of specific peptides depends on their capacity to be intact when they reach their target organs [56]. These peptides can also be assimilated by microorganisms (e.g., *Kluyveromyces marxianus*) and have also been used in the preparation of phosphopeptide complexes, which are important for the intestinal absorption of minerals [18].

Some of the bioactive peptides reported in the literature are the α- and β-lactophorin (derived, respectively, from α-lactalbumin and β-lactoglobulin), which have opioid agonist ACE inhibition, non-opioid stimulatory effect on ileum and ileum contraction, lactoferroxin (derived from lactoferrin), which act as an opioid antagonist and serophorin (derived from serum albumin) which have opioid activity and casoplatelins (derived from glycomacropeptide), which have antithrombotic activity [40,70]. The most studied and commercially available peptides are inhibitors of angiotensin I-converting enzyme (ACE) and antimicrobial agents. ACE inhibitory peptides, derived from α-lactalbumin and serum albumin, are effective in lowering hypertension and play a role in enhancing cardiovascular health, being highly beneficial in a wide range of bioactivities [42,43,56]. Fermentation with highly proteolytic strains of lactic acid bacteria is a successful strategy to produce antihypertensive peptides [56]. Most of the reports on ACE-inhibitory and/or antihypertensive peptides refer to peptides derived from bovine milk. However, in recent years, sheep’s and goat’s whey proteins have also become an important source of ACE-inhibitory peptides [51]. Other important peptide, lactoferricin, generated by the action of pepsin on lactoferrin has antimicrobial, antifungal, antiviral, antitumor and anti-inflammatory activities [42]. Dullius et al. reviewed the benefits of bioactive whey peptides available for food processing and compared their production processes and development in laboratory conditions as opposed to industrial production [41].

Hydrolysis of whey proteins has also been employed to modify the functional properties of proteins such as, solubility, viscosity, emulsifying and foaming properties, as well as to improve their nutritional properties [71].

In view of the nutraceutical and functional potential of SCW from buffalo’s milk processing, Sommella et al. studied the SCW profile. In this approach, the authors isolated a peptide fraction (obtained from α, β and k caseins), with molecular weights lower than 3 kDa and revealed a high complex profile using an LC-HRMS-based method. The peptidomic analysis indicated the presence of peptides with possible health benefits [6]. A study conducted by Monari et al. reported the enzymatic valorization of the protein fraction of SCW by means of UF with membrane cut-offs from 0.5 to 4 kDa. The protein-enriched fractions were used for the optimization of enzyme-based digests envisaging the production of potentially bioactive peptides [72].

Despite the information available for bioactive peptides resulting from whey hydrolysis, there is still a need to deepen the studies regarding the potential for bioactive peptide production from bovine and especially from caprine and ovine SCW. The work of Sommella et al. until now was the only available report [6].

## 5. Liquid Whey and Second Cheese Whey Concentrates

The production of liquid CW or SCW concentrates obtained by UF in small/medium size dairies can be a solution for the valorization of such by-products in view of their use as new ingredients in food product’s development. Liquid whey protein concentrates (LWPC) result from the selective concentration of whey proteins by means of UF, where these concentrates are used directly as ingredients in the manufacturing of other dairy products. Henriques et al. studied the effects of the use of LWPC partially replacing skimmed milk powder (SMP) in yoghurt formulations. The chemical composition of LWPC with a pH 6.35 was around 21.5% of total solids, of which, 8.60% was proteins, 7.10% was fat and 0.85% was ash [29]. The same authors also evaluated the gelation properties of LWPC produced by UF as raw material for thermally and acid induced gels intended for food applications. These applications allow for the production of highly nutritional dairy products and can be an economical alternative to the use of powdered products [73,74]. The acid-induced gels were produced with non-defatted LWPC by bacterial fermentation and by glucono-δ-lactone (GDL) acidification. The chemically acidified gels produced stronger gel structures than the equivalent fermented systems. The whey-based dairy gels obtained by fermentation or by the acidification promoted by glucono-δ-lactone presented viscoelastic behavior, appealing functional and nutritional properties, and their utilization can effectively contribute to the reduction of waste. Recently, Pires et al. also used LWPC from bovine whey to produce whey cheeses with added kefir or probiotics [12].

Depending on the composition of SCW and of the concentration factor applied in UF, liquid second cheese whey concentrates (LSCWC) produced can have 7–12% protein and variable levels of fat, lactose and minerals, depending on the process conditions applied. These LSCWC can be used in products such as salad dressings or fermented drinks allowing their on-plant recovery [27]. These authors studied the use of caprine LSCWC using two approaches. The first corresponds to the production of LSCWC from SCW through UF. The obtained LSCWC containing lactose and salts was used as a functional ingredient in the production of salad dressings. Besides, a desalted and lactose depleted LSCWC was also obtained by means of DF for the removal of lactose and salts. The resulting product contained about 10 times less lactose and salt than the ones of the product obtained by UF and was used as a functional ingredient in the production of fermented milk drinks. The processing scheme can be observed in Figure 3. The direct use of LSCWC in food products can be a good alternative for SCW valorization in medium and small companies since it does not require expensive equipment.

Borges et al. used LWPC, liquid buttermilk (LBM) and liquid sheep’s SCW concentrate as fat replacers in the production of reduced fat washed curd cheeses, envisaging the improvement of their flavor, texture and sensory properties. Those ingredients were incorporated into the milk, in the proportion of 5% (*v*/*v*). The reduced fat cheeses incorporating LWPC, LBM and LSCWC were compared to conventional reduced-fat cheeses and full-fat cheeses. Reduced fat cheeses with 5% incorporation of buttermilk presented the best results, both concerning texture parameters and sensory evaluation. However, the authors indicate that adequate mixtures of such by-products should be tested for similar purposes [75].

Another study reports about the use of cinnamon extract as an antimicrobial agent in the production of LWPC-based edible coating. The edible coating based on LWPC with cinnamon extract increased the shelf-life of fresh curds, increasing their functional value and contributing to a more sustainable production process [76].

Table 4 provides a summary of the main food/non-food applications of CW and SCW based products. In most food applications these products are used in the powder form (WPC, WPI or WPH). CW and SCW can also be used in liquid form, either directly when used as substrate for fermentation, or after a concentration step, as is the case of LWPC. The table is not exhaustive but highlights the main functions performed by the added ingredients.

## 6. Technologies Applied in the Valorization of Cheese Whey and Second Cheese Whey

The development of membrane separation techniques has been essential for the food industry that has taken advantage of their relatively easy scale-up, as well as to the fact of being inexpensive when compared with preparative chromatographic techniques. Membrane separation technics also offer the advantage that during processing the bioactive compounds do not suffer drastic heat treatments [10,58,59]. Several methods, such as UF, DF, NF, ion exchange chromatography, electrophoresis, crystallization and precipitation have been used to concentrate and separate proteins from other components present in whey. These techniques have been applied not only to the major compounds (β-lactoglobulin and α-lactalbumin), but also to other minor compounds, such as lactoperoxidase, lactoferrin, immunoglobulins, as well as other products such as the biologically active peptides obtained by enzymatic or chemical modification of proteins [43]. 

In the traditional process of whey demineralization, whey is concentrated by evaporation or reverse osmosis (RO) followed by demineralization of the concentrate using ion exchange columns. These processes are widely used at an industrial scale but imply high investment and operation costs [24]. 

For human food applications it is generally necessary to concentrate the whey protein. For the separation and purification of whey proteins, centrifugation and membrane technologies are the most used [40,83]. For the obtention of WPC or WPI, it is necessary to selectively concentrate the solid components and then further concentrate the product by RO or evaporation before dehydration. Membrane filtration has been important in concentrating desirable whey components and removing others. Equally important has also been the evolution of drying technologies [42].

Different membrane filtration techniques, such as microfiltration (MF), UF, NF, RO and electrodialysis are used to obtain WPC, WPI or WPH [45]. Besides these, for WPI production, adsorption techniques such as ion exchange are also used [83].

With the increasing evolution and utilization of membrane technologies in the dairy industry and their ease of operation, it is a possible, and can be an economic option, their application for the treatment of CW and SCW in small/medium scale dairy plants, in view of their use as ingredients for food products, such as salad dressings, cheese sauces or fermented drinks. UF membranes have low rejections (high permeability) for lactose, and salts (NaCl, KCl) and have high rejections of the nitrogen compounds and fat. Besides, the processing of UF permeates by NF has two major advantages. Firstly, the production of a clean effluent and the reduction of wastewater due to the possible reuse of NF permeates (e.g., as water used in cleaning processes). Secondly, the production of lactose concentrates with several potential industrial applications. The NF process has the advantage of simultaneously concentrating and demineralizing the whey, leading to a reduction of the total costs (equipment, energy) and the reduction of the wastewater disposal. 

In the mid-eighteenth century, membrane phenomena were observed and studied, mainly to evaluate barrier properties and related phenomena. The first commercial membranes for practical applications were manufactured by Sartorius in Germany after World War I, the know-how necessary to prepare these membranes originating from the early work of Zsigmondy [84]. The big step in the commercial applications of membrane filtration processes in the dairy industry occurred in the 1960s [34]. Nowadays, membrane processes are used in a wide range of applications and the number of applications is still growing [20,42,84]. In the food industry, a group of French researchers from INRA, started using membrane technologies to concentrate the milk used in cheesemaking. Membrane processing has allowed for uniform composition of the cheese milk and starter cultures activity has become more predictable [85]. After these first steps, the growth of these new technologies has been exponential and had a strong impact on the food industry and the agribusiness sectors. Several food products depend on membrane separation processes for their production. Examples of such products and applications are fruit and vegetable juices, wines and some grape products, the clarification of beer and the valorization of fish proteins [86]. However, the dairy industry can be considered as the main sector of the food industry benefiting from these technologies. 

Membrane separation processes are widely used to obtain proteins and lactose concentrates from CW and SCW [13,34,87]. These processes of tangential filtration consist of passing a liquid through a semi-permeable membrane. Two fractions are obtained: the retentate or concentrate, which is the portion containing molecules that cannot pass through the membrane, and the permeate or filtrate which is the fraction that crosses the membrane. Usually the different types of membrane are described with reference to their pore size or their cut-off point with respect to molecular mass [42]. Tangential filtration processes may employ five types of membranes, sometimes in combination: MF, UF, NF, RO and electrodialysis (ED) [45]. This technology presents some advantages when compared to traditional separation processes, namely, the reduction of wastewater production, the possibility of reuse and production of a clean effluent [34], low energy consumption when compared with evaporation and distillation processes, ease of combination with other separation processes and use of moderate process conditions, which is a very important issue in the food industry. Furthermore, additives and solvents are not required and the separation of components from a mixture is highly selective [10]. MF, UF, NF and RO have been reported by different authors [12,15,34,88,89] for valorization of CW and SCW, recovery of lactose and of protein fractions Figure 4. Membrane processes are normally followed by spray or freeze drying to obtain a dry (less than 5% moisture) product, and the combination of these processes is utilized for the production of whey powders with different protein contents. Table 5 presents a comparison of filtration techniques, comparing pore sizes and the type of components retained in each process when applied to milk [45].

The separation of whey components by UF was first done in 1971 [45]. UF has been the most widely used process to separate the protein fraction from the lactose fraction, the former being retained by the UF membranes, while the second constitutes most of the solid fraction of the permeate [91,92]. The UF permeate contains about 90% of the total whey solids. It is a source of lactose but also contains minerals and non-protein nitrogen compounds such as urea, free amino acids, creatine and creatinine, which are present in the initial whey and pass through the membrane [91,93].

Generally, to concentrate whey proteins, the molecular weight cut-off used is 10 kDa and UF is usually performed at temperatures below 50–55 °C, with a transmembrane pressure around 300–400 kPa and a membrane pore size of 250 nm. CW retentate contains protein, fat, insoluble salts, lactose and soluble minerals that did not cross the membrane, while the permeate contains mostly lactose [45]. The concentration of lactose and soluble salts in the retentate remains similar to that of the feed. The recovery of the components present in the UF permeate has been carried out by several processes, namely RO, which, more recently, has been replaced by NF. This process has been used by the dairy industry mainly to concentrate and demineralize acid whey and sweet whey. NF has an important function in the dairy industry for valorization of whey because it allows for protein concentration and partial demineralization of the product. Compared to RO, it provides an energy gain of approximately 45%, as it uses lower transmembrane pressures. When used as an alternative to evaporation (or RO) and electrodialysis to concentrate and demineralize the whey, energy and environmental costs can be significantly reduced [94]. 

Lactose is an important ingredient in infant formula and in food processing. The content of lactose in SCW is very high and its recovery may be a solution, which combined with the valorization of the nitrogen fraction, can be appealing. For lactose recuperation from dairy products evaporation, crystallization and spray drying are normally used. Lactose is mainly recovered from selective membrane separation technologies such as UF and NF. The permeate from the UF/NF processes containing around 4–8% lactose and 0.5% salt is concentrated in multiple effect evaporators. Lactose is then crystallized, separated and dried into lactose powder. To enhance the yield and purity, the UF permeate is also often processed by NF to partially remove salts and to increase the lactose concentration [39]. 

Thus, the sequence of UF/NF operations allows for the separation and recovery of the various whey fractions, predominantly the protein fraction and the fraction consisting mainly of lactose and salts excluding the water component.

Many different membranes for the UF process were studied. Macedo et al. compared the performance of three UF membranes used for SCW processing. In this study the authors compared cellulose regenerated acetate, composite fluoro polymer and polysulphone permanently hydrophilic membranes with similar MW cut-off (10 kDa). The results showed better performance for the cellulose regenerated acetate membrane because it had the highest permeate fluxes, the lowest irreversible fouling, good selectivity and the water recovery fluxes was around 100%. The higher hydrophilicity of regenerated cellulose membrane justified these results [28].

Monti et al. studied the chemical composition of bovine Ricotta cheese whey (scotta) of different origins and evaluated combined membrane technologies to separate and recover single constituents. The authors used a membrane fractionation system composed by two UF and one NF process for testing different combinations of membranes and operational parameters. The results showed the difficulty for the evaluation of the membrane differences because protein denaturation phenomena occurring during the production of Ricotta negatively influenced the separation and recover of chemical constituents. The principal problem was the fouling phenomena and protein–membrane interactions, that inhibited a complete separation between whey proteins and peptides. However, the authors refer that the UF process followed by NF allowed for the separation of one pure fraction including whey proteins and peptides, and of another with 80% of the original lactose concentrated. The membranes used in this study were based on Polyethersulfone (PESH), Polysulfone (PS) or Regenerated Cellulose (RC) for the UF process while for the NF Process one with Composite Thin Film was used [35].

Another study carried out by Pires et al., evaluated the incorporation of kefir and probiotics in Requeijão produced with whey concentrated by UF. In this work the concentration of bovine whey by UF allowed to produce bovine whey cheeses with advantages in comparison to traditional procedures. The approach allowed for a substantial reduction of the energy costs because of the reduction of 20 times the volume of whey being submitted to thermal processing. Whey cheese yield represented ca. 35% (*w*/*w*) of ultrafiltrate whey concentrate [12].

The most common problems associated with membrane processes are related to fouling or with the concentration polarization phenomena, which are responsible for the decline of the permeation fluxes, and for changing the selectivity of the processes [28,39]. These phenomena must be minimized due to their effect on reducing permeate flux and membrane selectivity, along with the costs of implementing cleaning cycles required to restore productivity [39].

Literature confirms that whey proteins and minerals, especially, calcium and phosphate, are the main contributors for the fouling of UF membranes [28,95]. Other components long-term affecting the performance of UF membranes are residues from processing, such as curd, residual lipids, caseinomacropeptide, enzymes and microorganisms [28]. These components probably adsorb onto the membrane surface, promote gelation in the polarized layer or induce pore blocking.

Fouling is very complex and depends on many physical and chemical factors such as concentration, temperature, pH, ionic strength and specific interactions. This occurrence not only reduces the flow, but also makes cleaning operations more difficult and expensive [28].

During the filtration process, the solution is transported to the membrane surface and due to its semipermeable nature, a portion of the solvent, with or without solutes, passes through the membrane. This causes a higher concentration of solutes on the membrane surface than in the solution, so some of these compounds return to it. The development of a concentration gradient of the components retained near the membrane is denominated concentration polarization. In particular, porous materials, such as MF and UF membranes, are highly susceptible to this type of fouling [39]. However, this problem is reversible, if the membrane is cleaned with water and appropriate cleaning solutions, the flow can be recovered if the process has not advanced [95]. 

## 7. Proposed Methodologies for Valorization of Second Cheese Whey (SCW)

Since most of the protein and fat present in whey are retained in whey cheeses, the residual proportion of these components in SCW, discourages its valorization. Therefore, it constitutes a serious environmental problem and its disposal constitutes a considerable expense for whey cheese making companies. For this reason, the transformation of SCW into useful products can be an interesting approach for the reduction of its environmental impact by allowing for its exploitation and valorization as a source of interesting compounds, similarly to what happened with whey in the last years. In this direction the European commission recommends the exploitation of by-products of the dairy industry as raw material for alternative processes [8].

Several alternatives for the recovery of SCW components can be applied. Some of which are: recovery of lactose to be used as sweetener in food products such as ice-creams, and baby food, ethanol fermentation, lactic acid or biogas [18,96]. Additionally, it has applications for the production of a biodegradable plastic component (polylactide, polymers, polyhydroxyalkanoates) [40].

Some studies have already been carried out for the use of the SCW (Table 6), but the information available is still scarce. As examples, we can refer the use of SCW for the production of fermented drinks with prebiotics and probiotics (symbiotic drinks) [8], for the production of lactose by crystallization [97], for lactic acid production [14] or for the obtention of bioethanol by yeast fermentation [16,22,37].

Maragkoudakis et al. demonstrated the capability of several lactic acid bacteria species to grow in pretreated SCW as substrate. The resulted products presented a high content of live beneficial bacteria and a low pH that favored product stability and hindered the development of potentially harmful bacteria. 

Regarding the production of dairy foods, Tirloni et al. studied the production of a ready-to-drink beverage produced from SCW. The shelf life of the product was determined at different temperatures and the potential thermal abuse was also investigated. The authors concluded that the addition of starter cultures seems promising and the addition of fruit puree was positive [33].

A study for lactose production using SCW and evaluation of the crystallization process at different pH levels and concentration factors was described by Pisponen et al. The results showed that the optimum acidity for lactose crystallization was close to pH 4.0, while at higher or lower pH levels the growth of crystals was inhibited. The crystal’s dimensions, the concentration factor and the qualitative properties corresponded to data provided in literature for lactose obtained from conventional cheese whey. The authors concluded that the results can be used to implement the crystallization process for manufacture of lactose from SCW [98].

Minhalma et al. used NF for the recovery of SCW organic nutrients. The SCW was processed by NF to recover the lactose fraction in the concentrate and a process water with a high salt content in the permeate. The permeation experiments were carried out with two NF membranes NFT50 (composed by a thin film composite on polypropylene) and the membrane an HR-95-PP (composed by a thin film composite on polyester). The NFT50 membrane showed the best results in terms of SCW fractionation and productivity. The results allowed the SCW fractionation into a salt depleted lactose concentrate that could be used as a raw material in the pharmaceutical, food or paper industries, and a salt enriched permeate almost free from organic matter [34].

Tsolcha et al. developed a biological (algal) study for SCW wastewater treatment system able to generate biodiesel while removing polluting nutrients and chemical oxygen demand (COD). The authors concluded that the well-adapted *Choricystis*-like algal could be efficiently used to treat SCW and the biomass produced could be harnessed as a source of biodiesel [99].

Kotoulas et al. studied the efficiency of natural zeolite to treat SCW and remove ammonium from artificial wastewater. The authors concluded that zeolite could retain a significant portion of nitrogen load from SCW and this by-product can be used as a fertilizing agent. It was concluded that the wastewaters treatment using zeolite can reduce pollutant loads and also recover significant portions of nutrients, which can be further reused [21].

Using a pilot-scale biological trickling filters in series with different operating conditions, Tatoulis et al. co-treated hexavalent chromium (Cr(VI)) with second SCW or winery effluents. The authors concluded that the wastewater could be used as a carbon source for Cr(VI) reduction. The use of two trickling filters in series could effectively treat wastewaters with very low installation and operational costs [100].

Zoppellari and Bardi explored the conversion of dairy effluents as renewable sources for bioethanol production. Different fermentation managements were tested to obtain the increased ethanol yields and process performances. Both CW and SCW showed to be suitable for bioethanol production [37].

Rama et al. reviewed CW and SCW reuse for biotechnological purposes. This review summarized literature on the use of both by-products as culture media for the growth of lactic acid bacteria (LAB), as cryoprotectants for freeze-drying and as encapsulating agents for the spray-drying of these microorganisms. CW was considered a good media for LAB growth and SCW also showed a good potential [11].

It was also demonstrated the feasibility of SCW fermentation to produce bio-ethanol by using *K. marxianus* [16].

Carota et al. studied the adequacy of SCW as a growth medium for lipid production. The authors used 18 strains of oleaginous yeasts to evaluate their growth and lipid-producing capabilities on this substrate. *C. laurentii* UCD 68-201, demonstrated to be a very promising candidate for biodiesel production using SCW as substrate [101].

Ribeiro et al. reported that the SCW has a very good potential to be used as a culture medium for *C. protothecoides.* Through an adequate stress strategy, it was possible to control carotenogenesis, allowing the production of high amounts of high value molecules [102].

Monari et al. used SCW from Ricotta cheese industrial production. After UF process the enriched protein fractions obtained were used in order to optimize enzyme-based valorization protocols [72].

Pereira et al. proposed the valorization of ovine cheese whey and SCW resulting from Portuguese manufacture of Requeijão, using thermocalcic precipitation and microfiltration (TP/MF) followed by ultrafiltration-diafiltration (UF/DF) to produce CW and SCW powders. The effect of TP/MF using two microfiltration membranes of 0.65 and 0.20 µm pore size on UF fluxes was evaluated by comparison with original CW and SCW UF fluxes. The authors concluded that clarification of by-products from ovine cheese manufacture by TP/MF significantly improved posterior UF treatments. The clarified products, as well as the MF retentates, were later used as ingredients on the manufacture of whey cheeses [15].

The same authors also studied the effects of the addition of WPC and clarification by-products obtained from ovine cheese whey and SCW on the yield and quality of the whey cheese. The addition of ovine WPC and clarification by-products on the manufacture of whey cheeses was considered interesting since it increased yield without affecting the gel strength of the products [36,101].

Membrane technology, namely NF was used for the recovery of SCW organic nutrients, resulting from Serpa cheese and curd cheese production. It was reported that the NF operation can reduce the wastewater organic load and simultaneously contribute to the valorization of the cheese and curd cheese manufacture by-products [97]. Table 6 summarizes the main conclusions of the reports regarding the valorization and characterization of second cheese whey.

**Table 6 foods-10-01067-t006:** Reports on the valorization and characterization of second cheese whey.

Applications	Techniques	Results	References
Valorize ovine whey and SCW by TP/MF for obtaining whey powders Evaluate the addition of WPC and clarification by-products obtained from ovine CW and SWC on the yield and quality of the whey cheese (Requeijão)	TP/MF and UF/DF UF	Clarification of by-products improved UF treatments. Increase in yield without affecting the strength of the products	[15,17] [36]
Profile of SCW from isolated peptide fraction	LC-HRMS-based method	Wide presence of valuable potential bioactive peptides	[6]
SCW used as substrate for production of a fermented probiotic drink	Microbiology	Good results for SCW as substrate for the production of a fermented probiotic drink	[8]
SCW used for lactose production	Crystallization	Good results obtained for the crystallization process for manufacture of lactose from SCW	[96]
Recovery of SCW organic nutrients	NF	SCW fractionation can be used as a raw material in the pharmaceutical, food or paper industries and minimize the wastewater environmental impact	[34,97]
Development of SCW wastewater treatment system for biodiesel and removing polluting nutrients	Microalgae	Algae could efficiently treat SCW and can be used for biodiesel production	[102]
Zeolite used to treat SCW and remove ammonium from artificial wastewater	Continuous flow column experiment	Zeolite nitrogen from SCW can be a fertilizing agent	[21]
Hexavalent chromium (Cr (VI)) was co-treated with SCW	Pilot-scale of biological trickling filters	Results indicate that the agro-industrial wastewater could be used as a carbon source for Cr (VI) reduction	[98]
Production of a ready to drink beverage produced from SCW with fruit puree	Culture addition	Addition of starter cultures was promising, and the addition of fruit puree improved sensory properties	[33]
Dairy effluents used to be converted in renewable sources for bioethanol production	Fermentation	Whey and SCW showed suitability for bioethanol production	[37]
Use of whey and SCW as media for the growth of LAB	Fermentation	Whey was considered a good media for LAB growth and SCW has a good potential too	[11]
Adequacy of SCW as a growth medium for lipid production	Fermentation	*C. laurentii* UCD 68-201, demonstrated to be a promising candidate for biodiesel production	[99]
SCW to be used as economic alternative substrate to grow microalgae	Fermentation	SCW has a very good potential to be used as a culture medium	[100]
SCW as a growth medium preserving biodiversity and maximizing bacterial cells concentration of natural starter cultures for pecorino Roman PDO cheese	Fermentation	A large concentration of cells was obtained in the modified SCW pellets, without modify the technological performance and microbial fingerprint.	[103]
Biogas production by anaerobic co-digestion of cattle slurry and CW	Anaerobic digestion	The mix has a similar energetic potential for anaerobic digestion as energy crops such as maize.	[104]
Fermentation of fruit-vegetable waste and CW for the production of H_2_	Fermentation	Considered a promising way for combining energy generation and lignocellulosic waste management.	[105]
Co-digestion of CW and glycerin	Anaerobic digestion	CW has great potential for methane production through anaerobic biological processes. However, it presents instabilities due to its high biodegradability. It is proposed its co-digestion with glycerin.	[106]

## 8. Conclusions and Future Perspectives

This work provides an overview about the existing literature regarding CW and SCW treatment processes and potential applications. The available strategies allow for reducing the environmental pollution and, above all, providing ways for further economic valorization of such by-products by their incorporation in food formulations. Although the issue has been for long object of research, recent developments and the need to transfer the available technologies to dairy companies, justify the work. Besides, the valorization of some dairy by-products is still neglected, as is the case of SCW.

The interest in CW and SCW is related to the presence in high percentages of whey protein, and other nitrogen components such as bioactive peptides, who have innumerous health benefits. Until now, the valorization of dairy by-products was focused only on CW, which already contributes to the development of new functional food ingredients, nutraceuticals and dietary supplements as well as products such as fermented drinks, whey cheese and yogurts. Other applications involve the preparation of culture media or biodiesel and bioethanol production. However, regarding SCW, only a few works address the valorization of this by-product.

Currently, the production of WPC, WPI and WPH, mainly from CW, offer the consumers these products in their simple form. However, recent developments envisaging the valorization of CW and SCW components, namely by the production of bioactive ingredients is boosting the development of new functional foods. 

In the case of SCW, future research should focus on resolving limitations in the use of these technologies to valorize it and in the benefits of the new SCW products to the human health.

Membrane technology processes can easily be implemented in dairy industry, even in small companies. In this way, research must be carried out in order to give to the dairy sector new solutions to enhance their by-products in order to increase their efficiency, economic gain, while at the same time, reducing costs with their disposal and prevent environmental pollution.

## Figures and Tables

**Figure 1 foods-10-01067-f001:**
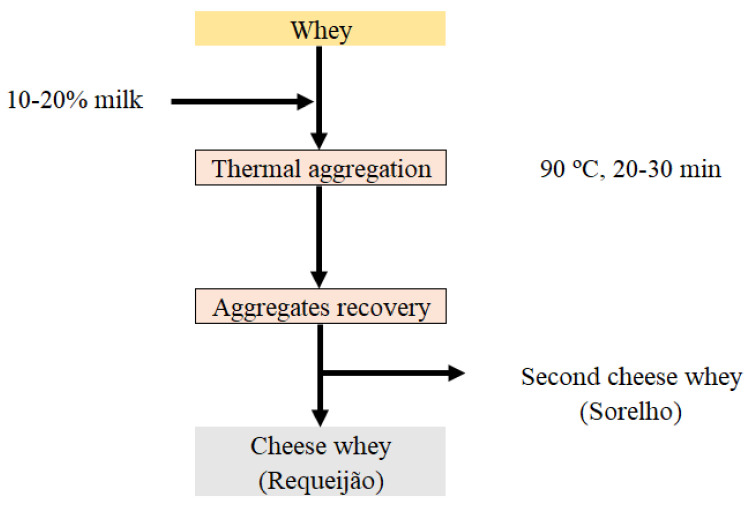
Scheme of whey cheese (Requeijão) manufacturing and second cheese whey (Sorelho) production.

**Figure 2 foods-10-01067-f002:**
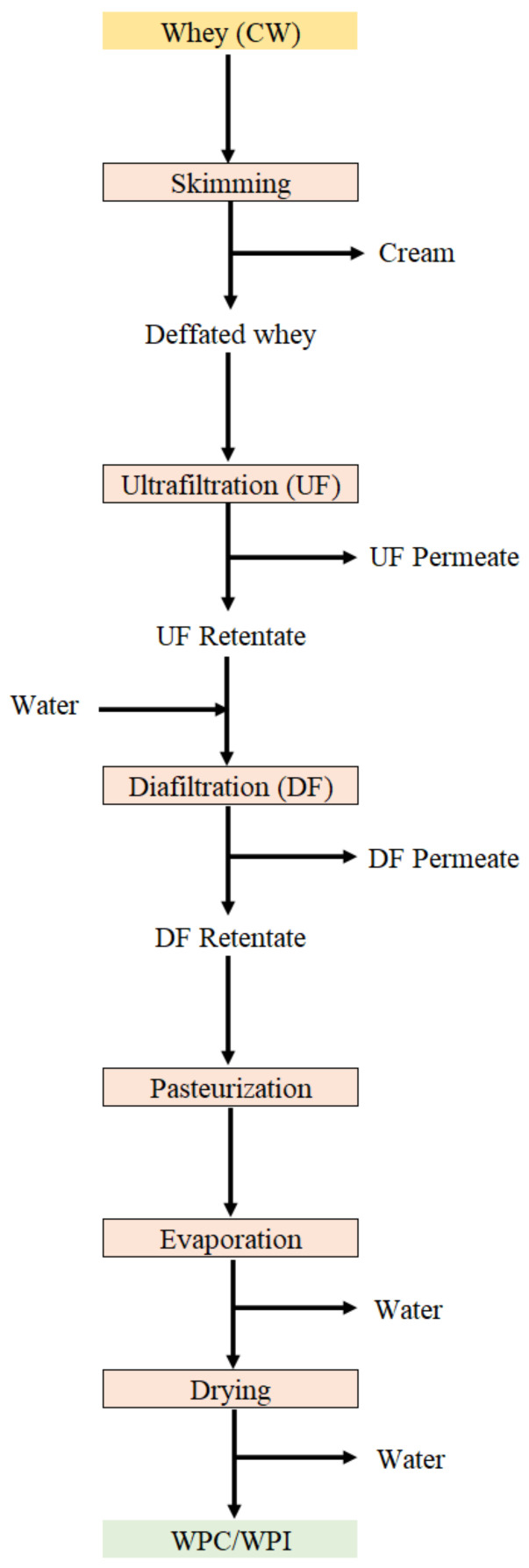
Representative scheme for obtaining whey protein concentrates (WPC) and whey protein isolate (WPI).

**Figure 3 foods-10-01067-f003:**
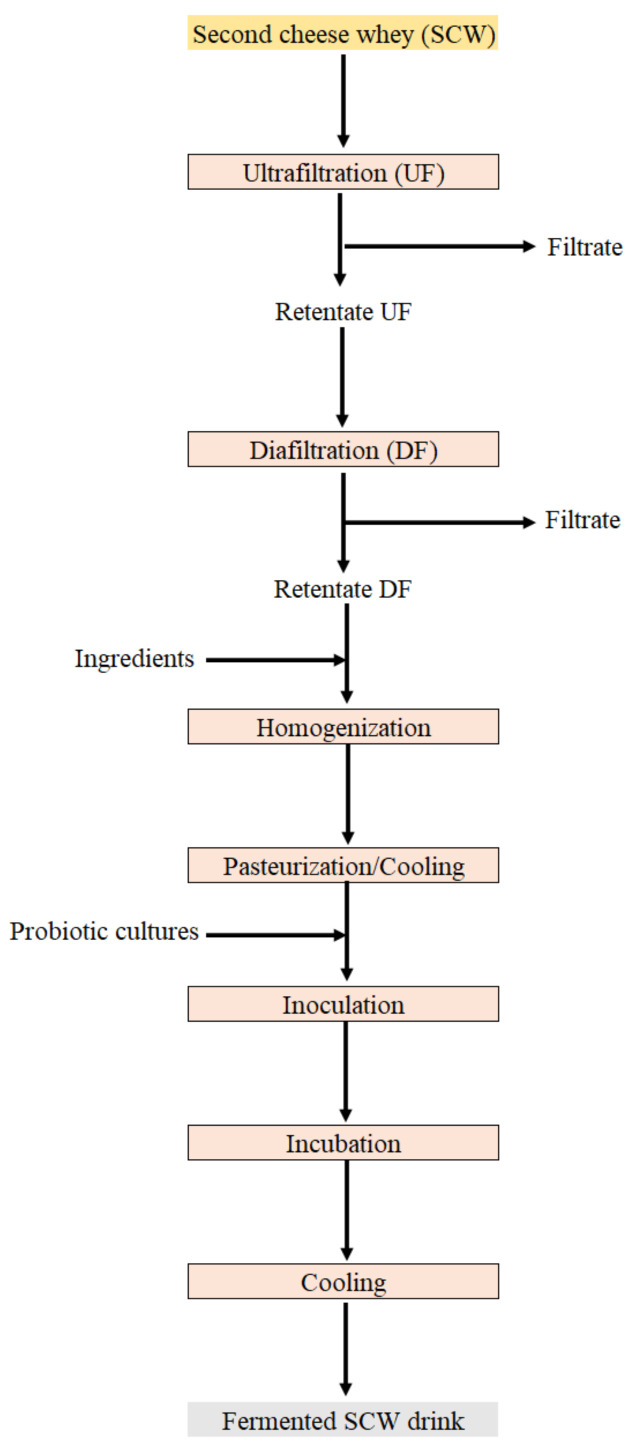
Example of UF/DF applied to SCW for the obtaining of fermented SCW drinks.

**Figure 4 foods-10-01067-f004:**
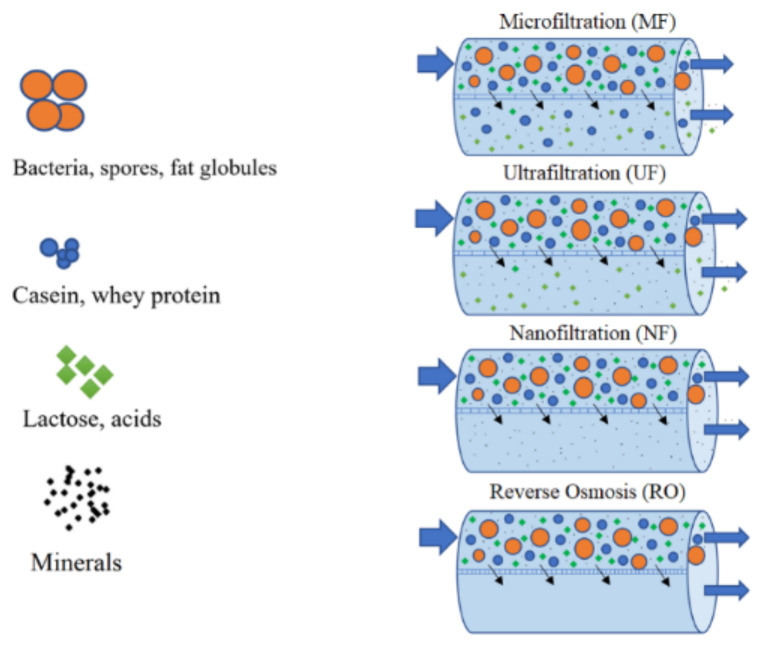
Membrane separation processes used to process CW and SCW.

**Table 1 foods-10-01067-t001:** Average composition of bovine, ovine and caprine whey and of ovine second cheese whey (% *w*/*v*).

	Bovine CW	Ovine CW	Caprine CW	Ovine SCW
Total solids	6.0–7.0	7.6–10.5	7.07–10.8	7.1–8.3
Lactose	4.2–5.0	4.3–6.1	5.02–6.7	4.5–5.7
Proteins	0.7–0.9	1.6–1.8	1.2–0.63	0.8–1.2
Fat	0.1–0.8	1.2–2.5	0.84	0.2–0.4
Minerals	0.5–0.6	1.0–1.8	0.57	1.7–1.9
pH	5.6–6.3	5.3–5.9	6.34	5.5–6.3

(CW = Cheese whey; SCW = Second cheese whey) Adapted from [28,29,30,31].

**Table 3 foods-10-01067-t003:** Identification of peptides of WPC hydrolyzed fraction.

Source Protein	Peptide Fragment	Amino Acid Sequence
α-La	f10–15	RELKDL
	f16–26	KGYGGVSLPEW
	f32–40	HTSGYDTQA
	f97–103	DKVGINY
	f97–104	DKVGINYW
	f98–104	KVGINYW
β-Lg	f33–42	DAQSAPLRVY
β-CN	f1–6	RELEEL
	f94–105	GVSKVKEAMAPK
CMP	f106–115	MAIPPKKNQD
	f107–115	AIPPKKNQD
	f161–169	TVQVTSTAV

(α-La = α-Lactalbumin; β-Lg = β-Lactoglobulin; β-CN = β-Casein; CMP = Caseinomacropeptide) Adapted from [58].

**Table 4 foods-10-01067-t004:** Examples of food and non-food applications of CW and SCW products.

APPLICATIONS	FUNCTIONS OF CW/SCW PRODUCTS
FOOD APPLICATIONS *DAIRY PRODUCTS*	
Reduced-fat/Low-fat cheese	Fat mimetics
Processed cheese	Emulsification/Water binding
Yoghurts/Low-fat yoghurts/Fermented drinks	Protein fortification/Fat mimetics
Ice-cream	Fat substitution/Emulsification/Foaming
*SAUCES/SALAD DRESSINGS/DRINKS*	
Sauces/Salad dressings	Emulsification/Fat mimetics/Creaminess
Drinks	Body/Creaminess/Viscosity
*MEAT PRODUCTS*	
Ham	Water binding/Gelification
Structured meat	Water binding/Gelification
Sausages and meat emulsions	Water binding/Emulsification/Fat mimetics/Gelification
*FISH PRODUCTS*	
Surimi	Water binding/Emulsification/Fat mimetics
*BAKERY AND CONFECTIONERY PRODUCTS*	
Bakery	Flavour/Egg substitution/Stabilization/Foaming
Snacks	Binding properties/Fat substitution/Foaming and expansion
*EDIBLE FILMS COATINGS*	
Edible films/coatings	Gas/Water vapour barrier
Edible films/coatings with incorporation biologically active components	Antimicrobials/Antioxidants
*DIETARY SUPLEMENTS*	
Infant formulae	Nutritional adequation
Elderly formulae	Nutritional adequation/Suplementation
Sport suplements (protein enriched bars, crackers and drinks	Nutritional suplementation
*FUNCTIONAL FOODS*	
Protein hydrolisates	Several health promoting functions (e.g., antihipertensive activity)
NON FOOD APPLICATIONS	
*BIOMEDICAL APPLICATIONS*	
Tissue engineering	Nanoparticles/Encapsulation
*FERMENTATION SUBSTRATES*	
For energy production	Algae/Lipids for biodiesel
For ethanol and lactic acid production	Ethanol; Lactic acid
For bioplastic production	Polylactide/polyhydroxyalkanoates
*ADHESIVES*	
Environmentally safe adhesives	Polymeriztion
*TEXTILES*	
Several applications in textiles	Enhanced staining/Abrasion resistence and tensile strenght/Flame retardancy/Antimicrobial properties/Microencapsulation of aroma

Sources [11,12,15,17,26,27,29,40,41,44,77,78,79,80,81,82].

**Table 5 foods-10-01067-t005:** Comparison of membrane separation of milk components.

Type	Pore Size (nm)	Retained Compounds	MW of Compound (kDa)
MF	20–4.000	Bacteria, fat globules and casein micelles	100–500
UF	20–200	Whey proteins	1–100
NF	<2	Lactose, divalent salts	0.1–1
RO	<2	Monovalent salts	<0.1
Electrodialysis	-	Removal of salt and deacidification of solutions containing neutral components	-
Pervaporation	-	Used for volatile organic pollutants	-

(MW = Molecular weight; MF = Microfiltration; UF = Ultrafiltration; NF = Nanofiltration; RO = Reverse osmosis) Adapted from [45,90].

## Data Availability

No new data were created or analyzed in this study. Data sharing is not applicable to this article.

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
