# Peer review of "Dairy By-Products: A Review on the Valorization of Whey and Second Cheese Whey"

_foods, 2021, doi:10.3390/foods10051067_

Round 1

Reviewer 1 Report

Whey utilization is one of the key issues of research and development in dairy science and technology. Whey is a typical by-product of dairy industry, its discharge present high environmental load, but it has a great potential for utilization because of its bioactive components. Utilization of whey components can answer the challenge of circular economy, as well. Therefore, the manuscript  foods-1207508 has an interesting topic that can provide useful information for the practice, as well.

The manuscript is generally well written with a logic structure. Introduction section is a good summary of general stat-of-art and research motivations. In section 2 the general characteristic of different whey are summarized. Section 3 provide information about the nitrogen compounds of whey. Section 4-5-6 focus on the process and technologies applied for whey or whey fraction utilization. Section 7 summarizes the SCW valorisation technologies. Tables and schematic diagrams are clear and meaningful. Manuscript contains interesting and valuable information and data which are based on relevant references. In my opinion the review manuscript provide useful and interesting information for the readers and it need just minor revision.

Comments and suggestions

I suggest the authors to give more details about the bioenergy utilization (non-food utilization) of whey.

I suggest the authors to give the whey based product in table, for instance.

From technological aspects, I suggest the authors to provide more information about the heat sensitivity of whey components.

Author Response

Thank you for the encouraging comments. Suggestions are appreciated and we altered the manuscript as proposed.

Two recent works regarding the impact of processing on whey products were introduced (references 64,65);

Examples of food and non-food applications were introduced in a new table (Table 5- new references 77-83);

More examples of bioenergy production were introduced in table 7 (references 105-107).

Reviewer 2 Report

This is a good review of the production, purification, and use of whey and its components. A small amount of reorganization is necessary to make it more readable.

Line 57: represents instead of means

Line 165: spreading instead of pulverization

Line 220: 5.1 instead of 5,1

Line 311: hydrolysates

Repetition: Lines 134-141, 299-306, 491-494. The information about membrane technologies should be contained in one section only. You can mention in line 134 that membrane technology will be discussed in section 6. Full names with abbreviations only have to be mentioned once. For example, you can write ultrafiltration (UF) the first time and UF thereafter.  

Author Response

Thank you for the encouraging comments. Suggestions are appreciated and we altered the manuscript as proposed.

Line 57: represents instead of means

Altered as proposed. Highlighted in yellow

Line 165: spreading instead of pulverization

Altered as proposed. Highlighted in yellow

Line 220: 5.1 instead of 5,1

Altered as proposed. Highlighted in yellow

Line 311: hydrolysates

Altered as proposed. Highlighted in yellow

Repetition: Lines 134-141, 299-306, 491-494.

The information about membrane technologies should be contained in one section only. You can mention in line 134 that membrane technology will be discussed in section 6. Full names with abbreviations only have to be mentioned once. For example, you can write ultrafiltration (UF) the first time and UF thereafter. 

Altered as proposed. Highlighted in yellow